# Stability Program in Dendritic Cell Vaccines: A “Real-World” Experience in the Immuno-Gene Therapy Factory of Romagna Cancer Center

**DOI:** 10.3390/vaccines10070999

**Published:** 2022-06-23

**Authors:** Elena Pancisi, Anna Maria Granato, Emanuela Scarpi, Laura Ridolfi, Silvia Carloni, Cinzia Moretti, Massimo Guidoboni, Francesco De Rosa, Sara Pignatta, Claudia Piccinini, Valentina Soldati, Luana Calabrò, Massimo Framarini, Monica Stefanelli, Jenny Bulgarelli, Marcella Tazzari, Francesca Fanini, Massimiliano Petrini

**Affiliations:** 1Osteoncology and Rare Tumors Center, Immunotherapy, Cell Therapy and Biobank, IRCCS Istituto Romagnolo per lo Studio dei Tumori (IRST) “Dino Amadori”, 47014 Meldola, Italy; annamaria.granato@irst.emr.it (A.M.G.); laura.ridolfi@irst.emr.it (L.R.); silvia.carloni@irst.emr.it (S.C.); massimo.guidoboni@irst.emr.it (M.G.); francesco.derosa@irst.emr.it (F.D.R.); sara.pignatta@irst.emr.it (S.P.); claudia.piccinini@irst.emr.it (C.P.); valentina.soldati@irst.emr.it (V.S.); luana.calabro@irst.emr.it (L.C.); monica.stefanelli@irst.emr.it (M.S.); jenny.bulgarelli@irst.emr.it (J.B.); marcella.tazzari@irst.emr.it (M.T.); francesca.fanini@irst.emr.it (F.F.); massimiliano.petrini@irst.emr.it (M.P.); 2Biostatistics and Clinical Trials Unit, IRCCS Istituto Romagnolo per lo Studio dei Tumori (IRST) “Dino Amadori”, 47014 Meldola, Italy; emanuela.scarpi@irst.emr.it; 3Immuno-Hematology and Transfusion Medicine, AUSL Romagna, 47121 Forlì, Italy; cinzia.moretti2@auslromagna.it; 4Oncologic and General Surgery, AUSL Romagna, 47121 Forlì, Italy; massimo.framarini@auslromagna.it

**Keywords:** immunotherapy, dendritic cell vaccine, quality control, ATMP, stability

## Abstract

Advanced therapy medical products (ATMPs) are rapidly growing as innovative medicines for the treatment of several diseases. Hence, the role of quality analytical tests to ensure consistent product safety and quality has become highly relevant. Several clinical trials involving dendritic cell (DC)-based vaccines for cancer treatment are ongoing at our institute. The DC-based vaccine is prepared via CD14+ monocyte differentiation. A fresh dose of 10 million DCs is administered to the patient, while the remaining DCs are aliquoted, frozen, and stored in nitrogen vapor for subsequent treatment doses. To evaluate the maintenance of quality parameters and to establish a shelf life of frozen vaccine aliquots, a stability program was developed. Several parameters of the DC final product at 0, 6, 12, 18, and 24 months were evaluated. Our results reveal that after 24 months of storage in nitrogen vapor, the cell viability is in a range between 82% and 99%, the expression of maturation markers remains inside the criteria for batch release, the sterility tests are compliant, and the cell costimulatory capacity unchanged. Thus, the data collected demonstrate that freezing and thawing do not perturb the DC vaccine product maintaining over time its functional and quality characteristics.

## 1. Introduction

In recent decades, ATMP trials have exponentially increased and applied to the treatment of various diseases, including cancer. The whole manufacturing process of ATMPs must be prepared in compliance with good manufacturing practices (GMP) to ensure the safety, quality, reproducibility, and efficacy of these therapeutic products [1]. The use of living cells in the final product formulation imposes some limitations on their use due to their rapid loss of viability and functionality. Within a GMP manufacturing process, the deemed preservation of the cellular products in quarantine until all sterility tests are completed, as well as the occurrence of a multiple-dose administration schedule, requires cryopreservation of the product for later use. To this aim, the recommendation of stability tests to evaluate the maintenance of drug quality parameters over time, and therefore to attribute a product expiration date, is becoming a requisite [2,3]. 

DCs represent the most important antigen-presenting cells (APCs), able to regulate innate and adaptive immunity through activation and tolerance. These key features highlight their central role in regulating immune response [4,5,6] and their extensive use in the context of cancer immunotherapies and clinical applications [7,8,9].

Since 2009, several clinical trials based on the DC vaccine have been ongoing at Immuno-Gene Therapy Factory of IRCCS-IRST Institute, which was licensed in 2012 by the Italian Medicines Agency (AIFA) for the production of autologous DCs pulsed with tumor homogenate [7].

The first fresh DC vaccine formulation is administered immediately to the patient, and the remaining DCs are cryopreserved for subsequent treatment administration. Therefore, a stability program in DC vaccines to address the issue of the conservation of quality parameters during the time in compliance with regulatory agencies [10] was developed. 

Currently, a variety of validated, reliable, and reproducible methods to assess the in vitro potency are available. For a long time, the potency of DC-based vaccines was evaluated by measuring their allostimulatory capacity in mixed lymphocyte reaction. However, this in vitro assay does not allow to separate the degree of stimulation due to the presentation of alloantigens from that induced by the costimulatory activity of DCs. Alternatively, DCs’ costimulatory activity could be indirectly assessed by the expression analysis of phenotypic markers [11] or using other assays such as the COSTIM and Co-Flow DC assay [12,13,14]. 

The assessment of the biological properties, defined as the ability or capacity of a product to achieve a specific biological effect, constitutes an essential step in characterizing an ATMP making use of reproducibility tests strongly encouraged by regulatory agencies.

In this report, we assessed the quality data of thawed DC final products in order to estimate their shelf life according to Food and Drug Administration (FDA) and International Council of Harmonization (ICH) guidelines. We performed the ELISPOT Costim assay as a potency test and evaluated the sterility, viability, and phenotype to demonstrate the stability and conformity to acceptance criteria at different time points upon cell thawing. 

## 2. Materials and Methods 

### 2.1. DC Vaccine Preparation

Mature DCs were achieved from ex vivo cultures of monocytes derived from peripheral blood mononuclear cells obtained by leukapheresis [15,16]. Briefly, monocytes were enriched in immature DCs with CellGro DC medium (Cell Genix, Freiburg, Germany) added with Interleukin-4 (Cell Genix, Germany) 1000 IU/mL and GM-CSF (Cell Genix, Germany) 1000 IU/mL. On day 6, at least 90% of the culture was pulsed with 100 μg/mL of autologous tumor homogenate, whereas the remaining was pulsed with Immucothel (Biosyn Arzneimittel, Fellbach, Germany) 50 μg/mL as an immunization control. After overnight incubation and eliminating the previous culture medium, pulsed immature DCs were cultured for an additional 2 days with a cytokine maturation cocktail compound of Interleukin-6, Interleukin-1β, Tumor Necrosis Factor-α (Cell Genix, Germany), and ProstinE2 (Pfizer, Latina, Italy or Cayman, Ann Arbor, MI, USA). On day 9, 10 × 10^6^ of DCs were harvested, washed, and resuspended in sterile saline for patient’s treatment (Figure 1a). The remaining DC aliquots were frozen in autologous plasma and 10% dimethyl sulfoxide (Mylan, Dublin, Ireland) by automated freezing (Planer Ltd, Middlesex, UK) and stored in nitrogen vapor (Figure 1b). 

### 2.2. Clinical Trials and Patients 

From 2013 to 2018, six patients’ batches were collected in the stability program. Three patients were treated in a compassionate use program, two in ABSIDE and one in ACDC clinical trial. For all patients, we obtained an exceeding number of DC vaccine doses than those required for treatment. The compassionate use program included metastatic melanoma patients treated in accordance with the Italian Ministerial Decree of 8 May 2003. According to this regulation, a drug can be requested for use outside clinical trial “when there is no valid therapeutic alternative to the treatment of serious illnesses, or rare diseases, or disease conditions that put the patient’s life at risk”. The ABSIDE clinical trial included melanoma metastatic patients treated with autologous DC vaccines combined with immunomodulating radiotherapy and/or preleukapheresis IFN-α in a randomized “proof-of-principle” phase II study (EudraCT number: 2012-001410-41). ACDC clinical trial included resected stage III and IV melanoma patients treated with autologous DC vaccines in a phase II randomized trial (EudraCT number: 2014-005123-27). All patient and batch characteristics are summarized in Table 1.

### 2.3. Thawing Time and Conditions

Frozen cells were thawed rapidly in a 37 °C water bath and immediately resuspended with 10 mL of sterile saline (Baxter, Rome, Italy). Cells were then centrifuged at 1000 rpm for 10 min; supernatant was removed, and cells were diluted in sterile saline at 1 × 10^6^ cells/mL. Cryopreserved mature DCs (mDCs) were thawed at different time points, 6, 12, 18, and 24 months, respectively. Furthermore, fresh DC aliquots were analyzed as baseline time points. 

### 2.4. Phenotypic Characterization of DC

mDCs were phenotypically characterized by flow cytometry. Cells were stained with the following mouse anti-human monoclonal antibodies (mAbs) labeled with fluorescein isothiocyanate (FITC) or phycoerythrin (PE): CD80 (BD Biosciences Cat# 340294, RRID:AB_2229132), CD86 (BD Biosciences Cat# 557343, RRID:AB_396651), HLA-DR (BD Biosciences Cat# 555561, RRID:AB_395943), CD83 (Beckman Coulter Cat# IM2410, RRID:AB_2335726) and analyzed by a FACS Canto flow cytometer (Becton Dickinson, Italy) equipped with a blue (488 nm) and red (635 nm) lasers. Appropriate isotype control antibodies were added to each analysis. Briefly, 3–5 × 10^5^ cells were suspended in 100 µL of PBS 1X and incubated for 30 min at 4 °C with a validated concentration of mAbs, then the cells were washed twice and resuspended in 400 µL of PBS 1X for the subsequent analysis. Ten thousand events were recorded for each sample, dead cells were excluded using a forward and side-scatter gate, and pulse geometry gating was used to remove doublets.

### 2.5. Viability Cell Counting by Trypan Blue Dye

Cells were counted by the Neubauer cell count chamber; viability was calculated by dividing viable cells by the total number of cells (live and not). 

### 2.6. Sterility

Sterility was evaluated for every thawed batch in accordance with European Pharmacopeia (EP 2.6.27 Microbiological Control of Cellular Product) using the BacT/Alert 3D Culture System (Biomerieux, Paris, France). The system is based on the colorimetric principle of detection of carbon dioxide produced by the contaminating microorganisms.

### 2.7. ELISPOT Costim Assay

ELISPOT Costim assay test was based on the Interferon Gamma (IFN-γ) ELISPOT method in which T lymphocytes were stimulated by DCs in the presence of a suboptimal amount of anti-CD3 antibody. Thus, the costimulatory capacity of DCs was measured by counting the number of spot-forming cells (SFCs) resulting from the IFN-γ secretion by activated T lymphocytes. The maintenance of DC costimulatory activity was assessed by comparing fresh DC vaccine with vaccine aliquots thawed every 6 months. CD3+ T cells co-cultured with DCs were obtained from three different healthy donors. PVDF membrane plates (Millipore, Milan, Italy) were activated with coating antibodies for human IFN-γ (U Cytech, Utrecht, The Netherlands). Subsequently, CD3+ T cells were plated at a concentration of 1 × 10^5^ cells/well in quadruplicate in serum-free CellGro DC medium. The same amount of CD3+ T cells were plated with 0.02 µg/mL of OKT3 antibody (negative control) or with 1 × 10^4^ DCs (Tcells:DCs ratio = 10:1) with and without 0.02 µg/mL of OKT3 antibody. ELISPOT plates were incubated for 22–24 h with 5% CO_2_ maintained at 37 °C. At the end of the incubation time, spots were revealed with detector antibody against human IFN-γ (U Cytech, Utrecht, The Netherlands) according to the manufacturer’s instructions and automatically read using ELIScan (AELVIS, Hannover, Germany). The costimulatory ability of DCs was measured by the mean of SFCs obtained after the subtraction of negative control SFCs. To ensure reproducibility, the same three reference T cell lots were used for every DC batch. 

### 2.8. Statistical Analysis

Descriptive statistics, including medians, standard deviations, and range, were used to analyze the viability and phenotypic markers HLA-DR, CD86, CD80, and CD83 measured at different time points. The differences between the medians were measured with a non-parametric Friedman test using statistical package IBM SPSS Statistics software version 26.

For the potency quality, the shelf life was evaluated according to FDA guidelines by linear regression and was analyzed by ANCOVA, considering time as the covariate. Slopes and time-zero intercept analysis for each batch was performed using 0.25 as the significance level, and regression analysis was performed on pooled data from all batches. Shelf life was then identified as the time until the mean potency of pooled cryopreserved products remained higher than the lower limit of the confidence interval at 95%, at an acceptance level of 70% of the potency observed at time zero (i.e., non cryopreserved product) and viability, phenotype, and sterility are keeping with acceptance criteria.

## 3. Results

### 3.1. Sterility

As a first step, sterility was tested by BacT/Alert 3D Culture System. All six batches were found to comply with the safety profile. 

### 3.2. Viability

We performed viability cell counting to evaluate the DC viability over time. The DC vaccines were thawed at several time points after 6, 12, 18, and 24 months, respectively. The cut-off limit for release was set to >70%. We analyzed the intra-batch viability in a time-dependent manner, and no significant difference was observed (Figure 2a). The viability percentage value of thawed aliquots gradually decreases over time with an absolute range between 82% and 99% but still remains greater than the defined acceptance criteria, as reported in Figure 2b. No significant difference was found for viability measured at different time points (Friedman test: *p* = 0.456). 

### 3.3. Phenotype

To evaluate if the level of expression of DC functional markers was impaired during the time, we characterized fresh and thawed DC vaccines for the expression of HLA-DR, CD86, CD80, and CD83. As shown in Figure 3, no significant difference was found for the expression of these markers measured at different time points (Friedman test: HLA-DR, *p* = 0.430; CD86, *p* = 0.619; CD80, *p* = 0.559; CD83, *p* = 0.129). All batches’ data analyzed met all acceptance criteria.

### 3.4. Stability of the Product Potency

In ELISPOT Costim assay, we tested six batches of DC-based vaccines at different time points. The response obtained from each DC batch after stimulation with three different donor T lymphocyte lots was evaluated, and we observed that the mean number of SFCs obtained was comparable between different allogeneic CD3+ responder cells (Figure 4a and Appendix A). All batches included in the stability program were evaluated in accordance with the provisions of the FDA guideline “Guidance for Industry Q1E Evaluation of Stability Data” (June 2004 ICH). Specifically, the “poolability” of all batches and the subsequent regression analysis on the overall data were performed. Data for poolability should not show statistically different slopes at the significance level of 0.25. The angular coefficient of the regression line indicates the change in immunological activity over time and must not be less than 70% vs. baseline. (Figure 4b). These results indicate that the cryopreservation for 24 months had no effect on the final in vitro DC potency. 

## 4. Discussion

DCs are important APCs and play a critical role in promoting the immune response. In the last decades, many studies have focused on the exploitation of these cells against cancer. Accordingly, DC vaccines have been in use for a long time with encouraging results [17,18,19,20,21]. The immunogenicity of DC vaccines has been established in most clinical studies; however, their clinical efficacy remains largely unexplored. In particular, the combination of DC formulations with currently approved adjuvant treatment options warrants further investigation [22,23]. Our cell factory was recently AIFA accredited to produce DCs according to GMP requirements through the validation of the manufacturing process. At the IRST-IRCCS cell factory, due to ethical concerns regarding patients’ submission to multiple invasive procedures (i.e., apheresis) and to minimize the manipulations of a large amount of raw materials, the first DCs dose is administered as freshly harvested, then cells are cryopreserved in ready to use aliquots for the subsequent doses. However, cryopreservation effects on therapeutic product properties are debated. Some reports suggest that freezing could reduce the functional properties of ATMPs, while others claim that cryopreserved products maintain functionality and characteristics comparable to that of fresh ones [24]. These conflicting observations on cryopreserved products make it mandatory to monitor the maintenance of quality parameters of thawed cell products over time and to define an appropriate product shelf life. 

In this study, we illustrate our IRST stability program where sterility, viability, phenotype, and potency of thawed products are assessed over time. Six DC vaccine batches were analyzed at the end of the manufacturing process (baseline) and at 6, 12, 18, and 24 months after freezing. The presence of viable cells was evaluated in the thawed products and confirmed that although the viability tends to decrease slightly as the cryopreservation time increases, no statistical difference was found between cryopreserved doses and fresh vaccines. These tests allowed the evaluation of the DC costimulatory capacity and the maintenance of our internal release parameters such as sterility, viability, and phenotype. 

The maximum decrease in the percentage of viable cells with respect to the fresh sample was 11%, and the minimum value of the percentage of viable cells was 83% observed in the batch thawed after two years of cryopreservation. All analyzed samples were compliant with the release criteria in terms of viability.

Furthermore, the phenotype of thawed products was investigated, and no statistical differences between the five time points were observed, meaning that the conditions of cryopreservation applied do not affect the expression of the phenotypic markers tested. 

The comparison between averages of IFN-γ SFCs over the time among a DC batch and three T lymphocyte donors demonstrated that freezing, storage, and subsequent thawing of cryopreserved vaccine aliquots do not significantly affect the costimulatory capacity of DC even after 24 months.

The variability of the potency data among different batches was probably linked to the complexity of the method or to the intrinsic biological activity of cellular product; thus, method standardization or eventual replacement with other validated tests, such as the Co-Flow DC [14], should be considered. This cytofluorimetric method could allow the evaluation of the absolute number of proliferating T cells in co-culture with mDCs and a suboptimal amount of anti-CD3 antibody and the accurate live cell seeding using Annexin V assay.

The analyses carried out over time confirmed that the DCs prepared by applying GMP guidelines and cryopreserved for two years in our facility are efficient and safe for patient administration. Moreover, these successful results allow the obtainment of a sufficient number of DCs from one leukapheresis for the coverage of the complete vaccination schedule. 

## 5. Conclusions

In conclusion, the followed stability program allows us to determine the expiry date and proper use of the DC vaccine. Statistical analysis performed on our data demonstrate that cryopreserved DC-based vaccines maintain their potency and functional capacity over 24 months after freezing. This represents an evident advantage for the safety of patients, the clinical protocol’s management, the reduced costs of production, and the quality of ATMPs product. However, further studies are needed to evaluate the functionality and phenotypic characteristics of the DC vaccine, and the future direction of our laboratory is to test the newly in-house-developed functional Co-Flow DC assay to answer this unmet need.

## Figures and Tables

**Figure 1 vaccines-10-00999-f001:**
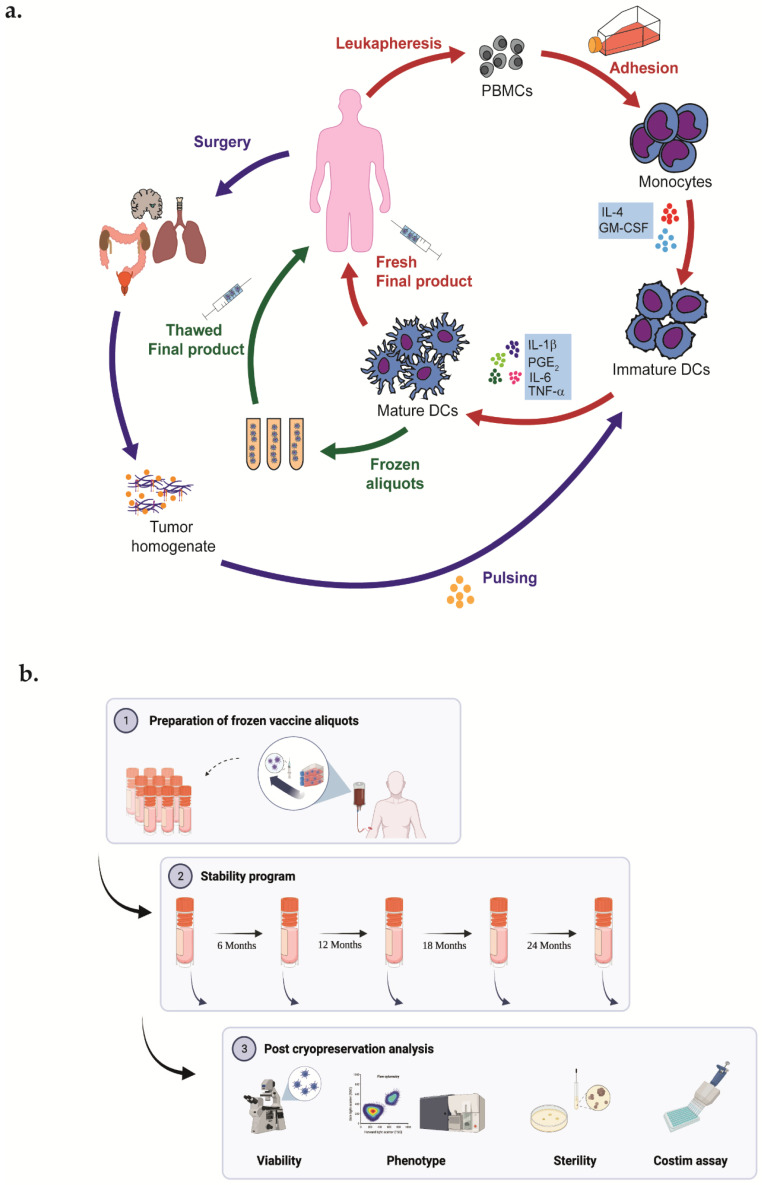
(**a**) Representative graphical of DCs vaccine manufacturing. Patients’ monocytes are obtained by leukapheresis procedure and subsequently differentiated into DCs. Cells are cultured with GM-CSF and IL-4; on day 6, DCs are pulsed with autologous tumor homogenate derived from surgically removed tumor; and on day 7 are matured for 48 h with TNFa, IL-1b, IL-6, and PGE2. Finally, a fresh final product is administered to the patient. (**b**) Representative graphical abstract of experimental plan of stability program created with Biorender.com.

**Figure 2 vaccines-10-00999-f002:**
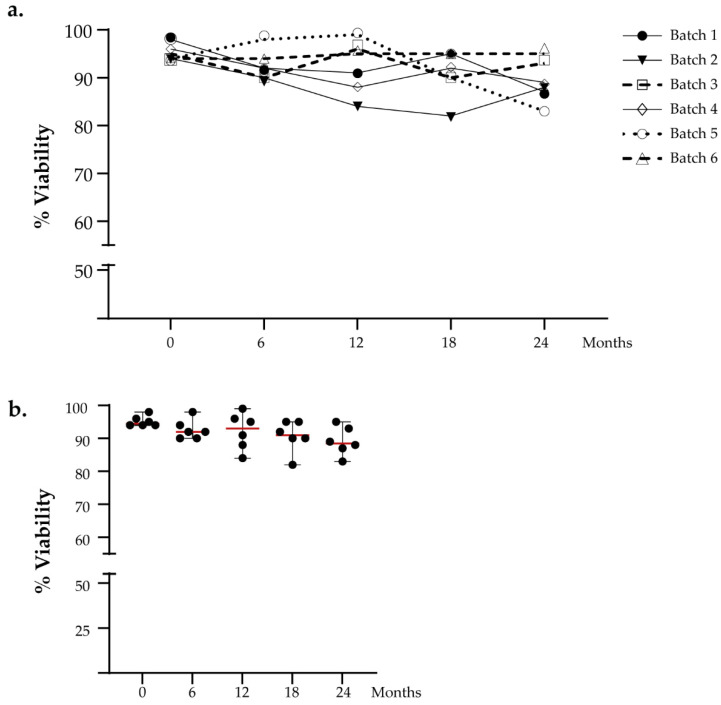
Functional testing of viable cell count. (**a**) The viability of DCs intra-batch was evaluated in a time-dependent manner. (**b**) The graph shows the viability analysis of all batches analyzed in the stability program. The red lines indicate the median value.

**Figure 3 vaccines-10-00999-f003:**
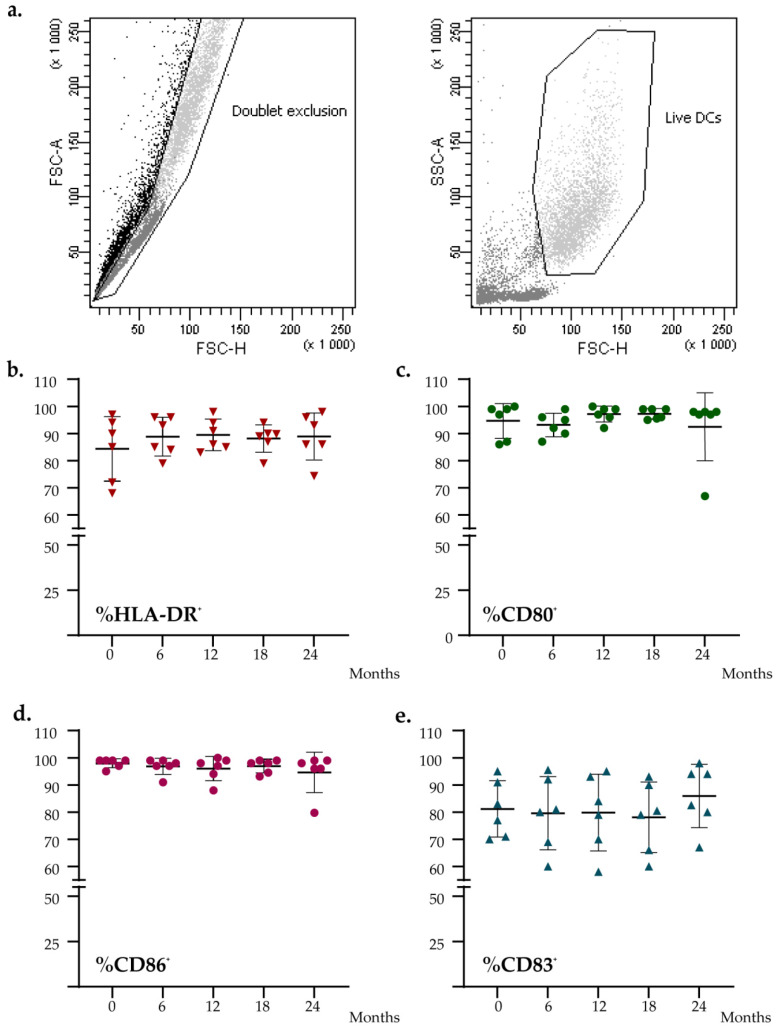
Flow cytometry analysis of the DCs vaccines at different time points. (**a**) Flow cytometry gating strategy. (**b**–**e**) Graph shows the percentage expression analysis of DCs maturation markers (HLA-DR, CD80, CD86, CD83). The black lines indicate the mean percent positive for each marker.

**Figure 4 vaccines-10-00999-f004:**
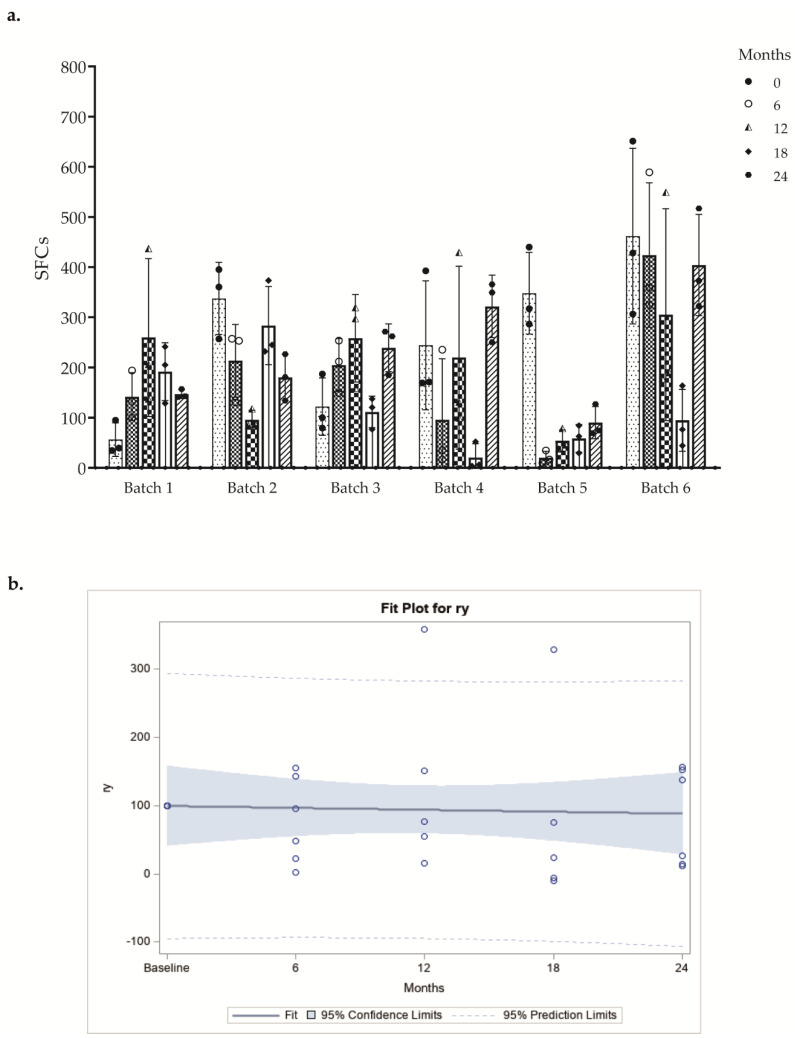
(**a**) Results of the ELISPOT Costim assay are shown as a bar-plot of SFCs (mean ± SD) at several time points. The different symbols shapes represent the mean of quadruplicate of each activated CD3+ T cell donor. (**b**) Regression analysis of poolability of all batches. Pooled regression line of the batches resulting after the tests of equality of slopes and intercepts of the regression lines. Each circle represents the mean value of the replicates of each donor at different time points.

**Table 1 vaccines-10-00999-t001:** Summary of patient and batch characteristics.

Batch Number	Sex	Age	Clinical Response	Clinical Trial	Cells per Vials (×10^6^)
1	M	73	PD	Compassionate use program	13.5
2	M	69	PD	Compassionate use program	15
3	M	63	CR	Compassionate use program	14.2
4	F	74	CR	ABSIDE	12.5
5	M	78	PD	ABSIDE	8
6	M	58	CR	ACDC	14.7

PD, progressive disease; CR, complete response.

## Data Availability

Data are contained within Appendix A.

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
