# Peer review of "Stability Program in Dendritic Cell Vaccines: A “Real-World” Experience in the Immuno-Gene Therapy Factory of Romagna Cancer Center"

_vaccines, 2022, doi:10.3390/vaccines10070999_

Round 1

Reviewer 1 Report

The authors have addressed my questions raised in the first review process.

Author Response

The authors have addressed my questions raised in the first review process.

Response: We would like to thank the reviewer for finding our manuscript of interest.

Reviewer 2 Report

Just one minor comment:

Section 2.7 Costim Assay, Result from One way Mix Lymphocyte reactions (MLR) will be more suitable here.  

Author Response

Just one minor comment:

Section 2.7 Costim Assay, Result from One way Mix Lymphocyte reactions (MLR) will be more suitable here. 

Response: We would like to thank the reviewer for his/her observation. Literature data shows that IFN- γ secretion is a well-known marker of antigen presenting cell costimulatory ability, therefore we decided to perform ELISPOT Costim assay and not one way MLR which output is cell proliferation. We would like to consider also proliferation assay to implement future study.

Reviewer 3 Report

Previous comments have only been partially addressed and current modifications actually raised more questions on the conclusion drawn,  in my opinion.

I still believe ELISPOT results should be showed in the context of the paper since these are the main support and basis for the conclusions presented.

In the description of the methods the authors mention that a suboptimal concentration of CD3 antibody has been used. Did the authors perform also a control with just CD3 antibody without any DC addition. Here the main question is whether DCs can still be T cell activators, so a negative control with just CD3 antibody is mandatory to exclude any contribution of CD3 stimulation that would mask any DC activity or loss of activity. 

Author Response

Previous comments have only been partially addressed and current modifications actually raised more questions on the conclusion drawn,  in my opinion.

I still believe ELISPOT results should be showed in the context of the paper since these are the main support and basis for the conclusions presented.

In the description of the methods the authors mention that a suboptimal concentration of CD3 antibody has been used. Did the authors perform also a control with just CD3 antibody without any DC addition. Here the main question is whether DCs can still be T cell activators, so a negative control with just CD3 antibody is mandatory to exclude any contribution of CD3 stimulation that would mask any DC activity or loss of activity. 

Response: We thank the reviewer for the comment and we have now added a better explanation of method in material and methods section (lines: 152-154 and 158-159). The ELISPOT costim assay were carried out with several control conditions, in particular CD3+ only, DCs + CD3+ and CD3+ + OKT3 0,02 µg/ml. The co-stimulation value is calculated by the difference between DCs + CD3++ OKT3 0,02 µg/ml SFCs and CD3+ + OKT3 0,02 µg/ml SFCs. We consider that showing raw data not fit the purpose of the article. Anyway, we have added the data statement section (lines: 291-293).

Reviewer 4 Report

Authors revised the text without additional data for further evaluation. Despite limited data,  authors replied the reviewer's comments.

It is better to adopt validated ELISpot assays to detect specificity of the immunotherapy, therefore concept of the cancer immunotherapy would lead to valuable impact.

Readers including me expect an article with scientific priority on the basis of valuable evidence.

Author Response

Authors revised the text without additional data for further evaluation. Despite limited data,  authors replied the reviewer's comments.

It is better to adopt validated ELISpot assays to detect specificity of the immunotherapy, therefore concept of the cancer immunotherapy would lead to valuable impact.

Readers including me expect an article with scientific priority on the basis of valuable evidence.

Response: We understand the reviewer’s concern about the ELISPOT costim assay; in fact, we speculate on validated assay for cancer immunotherapy developing a new validated method to evaluate cell potency (Carloni et al. PMID: 34072360). We have implemented our control quality analysis with the new functional Co-Flow DC assay to better define the DCs stimulatory efficacy for the prospective studies.

Reviewer 5 Report

Please provide the concentration of anti-CD3 ("suboptimal amount of anti-CD3 antibody") used for stimulation within the ELISPOT, thank you.

Author Response

Please provide the concentration of anti-CD3 ("suboptimal amount of anti-CD3 antibody") used for stimulation within the ELISPOT, thank you.

Response: We would like to thank the reviewer for finding our manuscript of interest. We have now added the information requested in material and methods section (Lines: 153-154).

Round 2

Reviewer 3 Report

Thank you to the authors for sharing the ELISPOT data which I think are important to support the manuscript conclusions.

I don't understand why the authors chose to show the data in a table format rather than in bar plots with error bars and statistical analysis reported.

I suggest to include the data in this presentation mode to enable a better understading of the data.

Author Response

Thank you to the authors for sharing the ELISPOT data which I think are important to support the manuscript conclusions.

I don't understand why the authors chose to show the data in a table format rather than in bar plots with error bars and statistical analysis reported.

I suggest to include the data in this presentation mode to enable a better understading of the data.

Response: We thank the reviewer for the comment and we have now added a better explanation of Costim ELISPOT showing the data as barplot graph (Supplementary Figure 1S). 

This manuscript is a resubmission of an earlier submission. The following is a list of the peer review reports and author responses from that submission.

Round 1

Reviewer 1 Report

In this manuscript, the authors carry out quality control testing of the impact freeze/thawing of patient derived dendritic cells has on their viability and functionality.  This study is part of ongoing clinical trials for the use of autologous dendritic cells in inducing anti-tumor immunity. The aims and study protocol are clearly written and presented thus making reading this manuscript straight forward.   The protocols used are appropriate and the conclusions reached by the authors are in accordance with the data generated.

Minor points:

  • Why is the viability cut-off set at 70%? Is there some underlying clinical readout/parameter that resulted in this cut-off point?
  • Figure 4: could the authors please show the actual data output for the functional testing of the dendritic cells with the ELISA spot assay? This is the most critical information regarding the functionality of the thawed cells and thus should be given more space than the poolability analysis.

Author Response

In this manuscript, the authors carry out quality control testing of the impact freeze/thawing of patient derived dendritic cells has on their viability and functionality.  This study is part of ongoing clinical trials for the use of autologous dendritic cells in inducing anti-tumor immunity. The aims and study protocol are clearly written and presented thus making reading this manuscript straight forward. The protocols used are appropriate and the conclusions reached by the authors are in accordance with the data generated.

Response: We would like to thank the reviewer for finding our manuscript of interest. Here below we provide responses to his/her comments.

Minor points:

  • Why is the viability cut-off set at 70%? Is there some underlying clinical readout/parameter that resulted in this cut-off point?

Response: We appreciate the reviewer’s discussion point. The viability cut-off has been set on the collection of our historical cell factory data that demonstrates that a value of cell viability lower than 70% was due to not optimal manufacturer procedure. Thus, the release criteria was set at 70% for cell viability as an indication of an adequate product quality.

  • Figure 4: could the authors please show the actual data output for the functional testing of the dendritic cells with the ELISA spot assay? This is the most critical information regarding the functionality of the thawed cells and thus should be given more space than the poolability analysis.

Response: We would like to thank the reviewer for his/her observation. To this aim, we have provided a better explanation of this assay (lines: 144-147) in the materials and methods section and we have now changed the title of section 3.4 (line: 201).

The scope of this article is focusing on the stability of cellular therapy products after freeze/thawing ATMP production procedures. Thus, testing poolability for design factors of stability studies is the most relevant test to determine product shelf lives (Appendix B of ICH Q1E guidance).

We consider that showing raw data not fit the purpose of the article. Anyway, we have added the data statement section (lines: 291-293).

Reviewer 2 Report

The authors summarized the stability of DC after different times stored in LN vapor, which is useful information for DC vaccines in future clinical trials.

Main Comments:

  1. Phenotypic characterization of DC.

CCR7 (CD197) should be included because it is critical for migration of DC to lymph nodes.

  1. Line 129, “100ml of PBS 1X” should be “100µl of PBS 1X”.
  2. Section 2.7 ELISPOT Costim Assay.  Since it was allogenic T cell that was cocultured with DC, the ‘empty-DC’ (non-antigen pulsing DC) was critical control for the assay. Do you have the data?
  3. Section 3.4. Potency.

Here the authors cited FDA guideline for data analysis, but the guideline was for stability data. Is it also specific for potency?

The author should give a detail description for how the data were analyzed. Correspondingly, there should be figure legend for Fig. 4.

If each circle represents the data from each experiment, according to the description in Materials and method, 6 batch of DC, 5 different time point, 3 different T cell lot, there should be 90 circles for data point from all experiments. But in the figure, there is only ~23 circles.  

Minor Issues:

  1. Please check grammar throughout the manuscript. For example, the title of the article should not end with a period (.).
  2. References: In a number of references, authors’ names are initials only, and/or missing page numbers or article number.

(1). Ref #5. P,S.? (2). Ref #7. O, G. et al?? (3). Ref #8. (4). Ref #15. (5). Ref #16. (6). Ref #15. Page numbers? (7). Ref #18. Article number? (8). Ref #25. Article number?

Author Response

The authors summarized the stability of DC after different times stored in LN vapor, which is useful information for DC vaccines in future clinical trials.

Response: We would like to thank the reviewer for finding our manuscript of interest. Here below we provide responses to his/her comments.

Main Comments:

  1. Phenotypic characterization of DC.

CCR7 (CD197) should be included because it is critical for migration of DC to lymph nodes.

Response: We appreciate the reviewer’s discussion point and we agree on the marker relevance for DC migration. In the past, we have focused on clinical migration evidence of our DC vaccine product using radioisotope-labeled DC; we demonstrated that a better migration activity to lymph nodes was obtained using intradermal than subcutaneous administration (Ridolfi et al., PMID: 15285807). Therefore, we implemented CCR7 (CD197) in our cytofluorimetric phenotyping analysis and we are evaluating the marker cut-off to define standardized criteria for the definition and validation of clinical results obtained.

  1. Line 129, “100ml of PBS 1X” should be “100µl of PBS 1X”.

Response: We thank the reviewer for the comment. We have now applied the correction (line: 128).

  1. Section 2.7 ELISPOT Costim Assay.  Since it was allogenic T cell that was cocultured with DC, the ‘empty-DC’ (non-antigen pulsing DC) was critical control for the assay. Do you have the data?

Response: We would like to thank the reviewer for his/her observation. To this aim, we have provided a better explanation of this assay (lines: 144-147) in the materials and methods section. The scope of this assay is testing the co-stimulatory ability of our DC product, regardless of any antigen specific stimulation which would require a specific batch of autologous T cells. Consequently, we do not consider useful introduce “empty DC” sample as a control in the assay.

  1. Section 3.4. Potency.

Here the authors cited FDA guideline for data analysis, but the guideline was for stability data. Is it also specific for potency?

The author should give a detail description for how the data were analyzed. Correspondingly, there should be figure legend for Fig. 4.

If each circle represents the data from each experiment, according to the description in Materials and method, 6 batch of DC, 5 different time point, 3 different T cell lot, there should be 90 circles for data point from all experiments. But in the figure, there is only ~23 circles.  

Response: We would like to thank the reviewer for his/her observation. We have now changed the title of section 3.4 (line: 201) and as suggested, we have added a legend to figure 4 to better describe data analysis (lines: 215-217).

According to ICH Q1E “Evaluation of Stability Data”, Guidance for Industry (FDA) guidelines, the shelf life was estimated based on the stability data of the batches.

The determinations were carried out in quadruplicate and the results were analyzed by evaluating for each condition the mean of the means of the replicate values obtained for each donor, excluding the replicates whose values were not included within the mean range +/- 1 standard deviation.

The regression line for each patient was plotted. Individual intercepts, individual slopes, and the pooled mean square error were calculated from all batches. As some of the shelf lives were shorter than that proposed, poolability tests was performed in order to determine whether the batches can be combined to estimate shelf life.

Before pooling the data from several batches to estimate a shelf life, a preliminary statistical test was performed to determine whether the regression lines from different batches have a common slope and a common time-zero intercept.

Analysis of covariance (ANCOVA) was employed considering the time as a covariate in order to test the differences in slopes and intercepts of the regression lines among batches at the significance level of 0.25.

The tests on the hypothesis of equality of slopes among batches showed p=0.380 and on the equality of intercepts was p=0.010, indicating that the data can be combined for the purpose of estimating the common slope. The shelf lives for individual batches in the stability study were estimated using the common slope and individual intercepts.

Minor Issues:

  1. Please check grammar throughout the manuscript. For example, the title of the article should not end with a period (.).

Response: We thank the reviewer for the comment. We have now checked grammar throughout the manuscript and applied grammar corrections.

  1. References: In a number of references, authors’ names are initials only, and/or missing page numbers or article number.

(1). Ref #5. P,S.? (2). Ref #7. O, G. et al?? (3). Ref #8. (4). Ref #15. (5). Ref #16. (6). Ref #15. Page numbers? (7). Ref #18. Article number? (8). Ref #25. Article number?

Response: We thank the reviewer for the comment. We have now modified references.

Reviewer 3 Report

English language should be extensively revised. Abbreviations such as "doesn't", "isn't" are not appropriate in a scientific publication. English grammar and expression should be reviewed by a mother tongue (e.g. line 66: "to achieved" should be "to achieve" etc.).

The quality and the scientific content are extremely poor, key details of  the experimental procedure and results were not specified; in particular: what antigens were used in the ELISPOT re-challenging assay ? Why the results from the ELISPOT analyses (which would be key in claiming the conclusion that DC maintain their immunogenic potential over time) are not reported in the manuscript? 

I suggest the rejection of the manuscript in its current form for publication.

Author Response

English language should be extensively revised. Abbreviations such as "doesn't", "isn't" are not appropriate in a scientific publication. English grammar and expression should be reviewed by a mother tongue (e.g. line 66: "to achieved" should be "to achieve" etc.).

Response: We thank the reviewer for the comment. We have now checked grammar and English expression throughout the manuscript and applied grammar corrections.

The quality and the scientific content are extremely poor, key details of  the experimental procedure and results were not specified; in particular: what antigens were used in the ELISPOT re-challenging assay ? Why the results from the ELISPOT analyses (which would be key in claiming the conclusion that DC maintain their immunogenic potential over time) are not reported in the manuscript? I suggest the rejection of the manuscript in its current form for publication.

Response: We understand the reviewer’s concern about the quality and the scientific content; however, we had to contextualize our data on the scenario of Advanced therapy medicinal products (ATMP). The quality requirements, such as stability in terms of viability, phenotype, and potency, after freeze/thawing ATMP production procedures are key factors to avoid repeated invasive procedures on patients ( e.i, apheresis ) and facilitate patient’s recruiting , to reduce the number of complex hand-operations, to obtain robust clinical trials data, to analyze robust historical data, to control production data,  to increase the sharing information about biological ATMP characteristic and to plan future clinical trial taking to account also logistic requirements (e.i,. ATMP administration in site or hub center).

We have now introduced a detailed information of the experimental procedure in Figure 1b (lines: 95-96; Figure 1b).

The scope of ELISPOT costim assay is testing the co-stimulatory ability of our DC product, regardless any antigen specific stimulation which would required specific batch of autologous T cell. Consequently, we did not test several tumor antigens. We have provided a better explanation of this assay (lines: 144-147).

We consider that showing raw data not fit the purpose of the article. Anyway, we have added the data statement section (lines: 291-293).

Reviewer 4 Report

Authors described DC vaccines being met the quality verifications based on the viability, phenotype, and potency using ELISpot assays. 

The DC vaccines cryoreserved under liquid nitrogen are required for safety and efficacy of clinical trials. I was wondering whether such validated DC vaccines could induce immunogenicity for the melanoma patients. It would be more interesting that authors provide the data based on the clinical response applied with the tumor-lysate pulsed DC vaccination protocol.

When ELISpot assays applied for the tumor-derived antigens, authors need to compare a tumor-lysate pulsed DC vaccine with the unloaded DC vaccine to determine the specificity.

Author Response

Authors described DC vaccines being met the quality verifications based on the viability, phenotype, and potency using ELISpot assays. 

The DC vaccines cryopreserved under liquid nitrogen are required for safety and efficacy of clinical trials. I was wondering whether such validated DC vaccines could induce immunogenicity for the melanoma patients. It would be more interesting that authors provide the data based on the clinical response applied with the tumor-lysate pulsed DC vaccination protocol.

Response: We would like to thank the reviewer for finding our manuscript of interest. We appreciate the reviewer’s discussion point. The clinical response of retrospective data was previously reported in the clinical article (De Rosa et al. PMID: 28654547) with encouraged outcome; however we could not publish data derived from clinical studies not yet completed. In this article, we focused on the stability of cellular therapy products in the scenario of Advanced therapy medicinal products (ATMP), in fact the quality requirements, such as stability in terms of viability, phenotype, and potency, after freeze/thawing ATMP production procedures are key factors to obtain robust clinical data.

When ELISpot assays applied for the tumor-derived antigens, authors need to compare a tumor-lysate pulsed DC vaccine with the unloaded DC vaccine to determine the specificity.

Response: We would like to thank the reviewer for his/her observation. To this aim, we have provided a better explanation of this assay (lines: 144-147) in materials and methods session.  The scope of this assay is testing the co-stimulatory ability of our DC product, regardless any antigen specific stimulation which would required specific batch of autologous T cell. Consequently, we do not consider useful introduce in the assay “empty DC” sample as control. 

Reviewer 5 Report

Pancisi and colleagues evaluated the stability of six dendritic cell (DC) preparations, which were differentiated from CD14+ monocytes using IL-4 and GM-CSF, stimulated with autologous tumor homogenate or immunocyanin as control, matured with a cocktail of IL-6, IL-1ß, TNF-alpha and prostaglandin E2, resuspended in autologous plasma plus 10% DMSO, and then stored in nitrogen vapor for 0-6-12-18-24 months. The DC preparations were obtained from patients suffering from metastatic melanoma with complete remission (n=3) or progressive disease (n=3).

The DC preparations proved to be sterile. The viability of the thawed cells measured by trypan blue staining ranged between 82-99%, with a slight drop over time. The phenotype of the DCs as evaluated by HLA-DR, CD80, CD83 and CD86 expression remained similar over time. In addition, the authors evaluated the “potency” of the DCs by co-culture with CD3+ T cells of three healthy donors and then measuring the IFN-gamma production in the absence and presence of anti-CD3 via ELISPOT. The authors conclude that the DC preparations remain sterile, viable, consistent in phenotype and potent over 24 months of storage.

The authors present a real-life analysis of the stability and quality of frozen DC preparations. The long periods of 18 and 24 months are worth reporting.

Major point:

  • Overall, the data are clearly presented, except for the ELISPOT assay. The authors need to find a better way than Figure 4 to show the raw data. In the results section, the authors say that “the mean number of SFCs obtained was comparable between different allogeneic CD3+ responder cells (data not shown)”, while in the discussion, they describe a “wide distribution of the mean number of IFN-gamma SFC”. It is not clear what is meant by “Fit Plot for ry” and “poolability of all batches”. In addition, please show controls without anti-CD3 as wells as controls without DC.

Minor points:

  • English wording and spelling needs to be checked.
  • Please describe Coflow DC in more detail.
  • Phenotypic characterization of DC: did the authors use PBS or DPBS? In addition, please show gating.
  • Discussion: what is meant by “minus value of percentage was 83%”?

Author Response

Pancisi and colleagues evaluated the stability of six dendritic cell (DC) preparations, which were differentiated from CD14+ monocytes using IL-4 and GM-CSF, stimulated with autologous tumor homogenate or immunocyanin as control, matured with a cocktail of IL-6, IL-1ß, TNF-alpha and prostaglandin E2, resuspended in autologous plasma plus 10% DMSO, and then stored in nitrogen vapor for 0-6-12-18-24 months. The DC preparations were obtained from patients suffering from metastatic melanoma with complete remission (n=3) or progressive disease (n=3).

The DC preparations proved to be sterile. The viability of the thawed cells measured by trypan blue staining ranged between 82-99%, with a slight drop over time. The phenotype of the DCs as evaluated by HLA-DR, CD80, CD83 and CD86 expression remained similar over time. In addition, the authors evaluated the “potency” of the DCs by co-culture with CD3+ T cells of three healthy donors and then measuring the IFN-gamma production in the absence and presence of anti-CD3 via ELISPOT. The authors conclude that the DC preparations remain sterile, viable, consistent in phenotype and potent over 24 months of storage.

The authors present a real-life analysis of the stability and quality of frozen DC preparations. The long periods of 18 and 24 months are worth reporting.

Response: We would like to thank the reviewer for finding our manuscript of interest. Here below we provide responses to his/her comments.

Major point:

Overall, the data are clearly presented, except for the ELISPOT assay. The authors need to find a better way than Figure 4 to show the raw data. In the results section, the authors say that “the mean number of SFCs obtained was comparable between different allogeneic CD3+ responder cells (data not shown)”, while in the discussion, they describe a “wide distribution of the mean number of IFN-gamma SFC”. It is not clear what is meant by “Fit Plot for ry” and “poolability of all batches”. In addition, please show controls without anti-CD3 as wells as controls without DC.

Response: We appreciate the reviewer’s suggestion. We have now added more detailed information regarding potency results (lines: 251-260) and improved figure 4 adding a legend to better describe data analysis (lines: 215-217).

According to ICH Q1E “Evaluation of Stability Data”, Guidance for Industry (FDA) guidelines, the shelf life was estimated based on the stability data of the batches.

The determinations were carried out in quadruplicate and the results were analyzed by evaluating for each condition the mean of the means of the replicate values obtained for each donor, excluding the replicates whose values were not included within the mean range +/- 1 standard deviation.

The regression line for each patient was plotted. Individual intercepts, individual slopes, and the pooled mean square error were calculated from all batches. As some of the shelf lives were shorter than that proposed, poolability tests was performed in order to determine whether the batches can be combined to estimate shelf life.

Before pooling the data from several batches to estimate a shelf life, a preliminary statistical test was performed to determine whether the regression lines from different batches have a common slope and a common time-zero intercept.

Analysis of covariance (ANCOVA) was employed considering the time as a covariate in order to test the differences in slopes and intercepts of the regression lines among batches at the significance level of 0.25.

The tests on the hypothesis of equality of slopes among batches showed p=0.380 and on the equality of intercepts was p=0.010, indicating that the data can be combined for the purpose of estimating the common slope. The shelf lives for individual batches in the stability study were estimated using the common slope and individual intercepts.

We consider that showing raw data not fit the purpose of the article. Anyway, we have added the data statement section (lines: 291-293).

Minor points:

English wording and spelling needs to be checked.

Response: We thank the reviewer for the comment. We have now checked grammar throughout the manuscript and applied grammar corrections.

Please describe Coflow DC in more detail.

Response: We appreciate the reviewer’s suggestion and have now added more detailed information regarding CoFlow DC (lines: 260-262).

Phenotypic characterization of DC: did the authors use PBS or DPBS? In addition, please show gating.

Response: We appreciate the reviewer’s suggestion and have now added more detailed information regarding gating strategy in the method section and in figure 3 of the manuscript (lines: 130 -131; Figure 3a). The reagent used for phenotypic characterization of DC was PBS (line: 128).

Discussion: what is meant by “minus value of percentage was 83%”?

Response: We thank the reviewer for the comment. We have now clarified the point by replacing “minus” with “minimum” (line: 245).